# Mechanism of client loading from BiP to Grp94 and its disruption by select inhibitors

Tara P. Azam, Jiaqi Han, Erin E. Deans, Bin Huang, Reyal Hoxie, Larry J. Friedman, Jeff Gelles ⓘ & Timothy O. Street ⓘ ✉

Hsp90 chaperones are a long-standing cancer drug target with numerous ATP-competitive inhibitors in clinical trials. Client proteins are transferred from Hsp70 to Hsp90 in a stepwise process of client delivery, loading, and trapping, but little is known about how inhibitors influence these steps. By examining the ER-resident BiP/Grp94 system (Hsp70/Hsp90 paralogs), we discover that some inhibitors allow BiP to push Grp94 into the client loading conformation, whereas other inhibitors block this conformational change and destabilize a BiP/client/Grp94 ternary complex. We uncover how BiP drives Grp94 into the client loading state and identify a structural explanation for why only a select group of inhibitors disrupt client loading on Grp94. These results show a client loading mechanism with specific shared features between the Hsp70/Hsp90 systems in the ER and cytosol and open a new avenue for rational Hsp90 drug design.

Hsp70 and Hsp90 are ancient chaperone families that maintain the activity and folding of their "client" proteins[1]. ATP-competitive Hsp90 inhibitors have been extensively investigated as anti-cancer drugs because many oncogenic proteins are functionally dependent on Hsp90[2]. The Hsp90 system in the cytosol was the first intended target of inhibitors in clinical trials[3], but the organelle-specific paralogs, such as Grp94 in ER, are now also drug targets[4].

While Hsp70 and Hsp90 family members can work independently, significant progress has been made in understanding how these chaperones can also work as a coordinated system[5,6]. Hsp90 receives clients from Hsp70 in a process that can be divided into steps of delivery, loading, and trapping (depicted in Fig. 1 for ER-resident paralogs BiP/Grp94). In client delivery, the Hsp90 dimer arms are open and Hsp70 can directly bind Hsp90 at "interface I", which is conserved from *E. coli*[7,8] to yeast[9] to humans[10] and for Grp94[11]. Subsequent ATP-dependent closure of Hsp90 allows the client to be stably trapped between the Hsp90 dimer arms[12]. Most Hsp90 inhibitors can bind Hsp90 in the open conformation and prevent closure by blocking ATP while also being sterically incompatible with the closed conformation (Supplementary Fig. 1). In contrast to the known effects of inhibitors on Hsp90 closure, little is known about how Hsp90 inhibitors impact client loading. Because Hsp90 inhibitors have been structurally characterized with the isolated Hsp90 N-terminal domain (NTD), many

questions remain about how inhibitors impact the structure and dynamics of the full-length Hsp90 dimer in the context of client proteins.

Hsp90 inhibitors exert particularly strong anti-cancer effects in cell lines where Hsp90 is found in stable assemblies with Hsp70 and other co-chaperones[13], suggesting that the combined Hsp70/Hsp90 system is an important biological target of inhibitors. It is unknown how inhibitors impact the structure and function of a combined Hsp70/Hsp90 pair, in part because the cytosolic Hsp70/Hsp90 system has a high level of conformational heterogeneity[14,15] and compositional heterogeneity from co-chaperones[16]. Here we show that the more pared-down BiP/Grp94 system, with its minimal co-chaperone involvement[17], provides a tractable system for examining the impact of Hsp90 inhibitors.

Like other Hsp90 family members, Grp94 undergoes an open-closed-open conformational cycle. ATP binding at the Grp94 NTD results in dimer arm closure[18]. Grp94 has a longer-lived closed state compared to other Hsp90 family members, which has enabled distinct steps of the Grp94 conformational cycle to be visualized in real time by single molecule FRET (smFRET)[19]. This method previously showed that BiP accelerates ATP-dependent Grp94 closure by stabilizing a partially closed conformational intermediate (termed the C' state)[20]. These previous findings left open the question of how BiP stabilizes the

Department of Biochemistry at Brandeis University, Waltham, MA 02453, USA. ✉ e-mail: tstreet@brandeis.edu

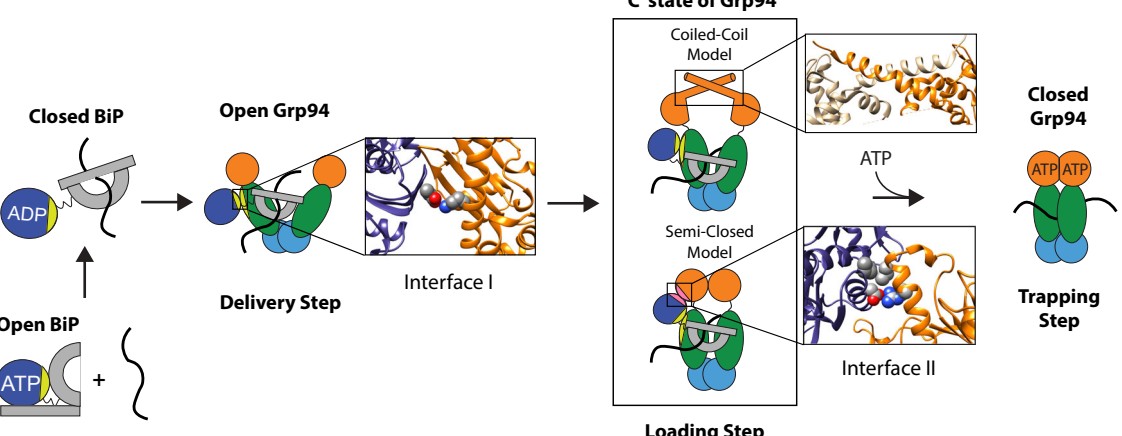

**Fig. 1 | Overview of key steps in client transfer.** In the ATP conformation, the BiP SBD (gray) and BiP NBD (navy blue) are docked and the SBD lid (gray rectangle) is open. Upon ATP hydrolysis, SBD lid closure can trap a client protein (black squiggle). In delivery, BiP binds Grp94 in the open conformation via interface I[11] (light green). Grp94 NTD is in orange, Grp94 MD is in dark green, and Grp94 CTD is in light blue. The inset highlights a conserved salt bridge at interface I (Hsp70$_{E218}$:Hsp90$_{K419}$ from PDB: 7KW7). For client loading, there are two candidate structural models for the Grp94 C′ state, based on the coiled-coil structure[21] (PDB: 5F3K) and the semi-closed structure[10] (PDB: 7KW7) of Hsp90. In the semi-closed structure, interface II (pink) is formed. The lower inset highlights key residues at interface II in the semi-closed structure of Hsp70/Hsp90 (PDB: 7KW7). The upper inset highlights the crossing of N-terminal α-helices in the coiled-coil structure. In the last step, Grp94 traps the client protein upon ATP-dependent closure.

Grp94 C′ state. Here we show that BiP stabilizes the Grp94 C′ conformation via contacts at a secondary interaction interface between the BiP nucleotide binding domain (NBD) and the Grp94 NTD. We also show that contacts at the secondary interface provide the driving force by which BiP accelerates ATP-dependent closure of Grp94. While the structure of the Grp94 C′ state is not known, this state can be trapped with a designed disulfide bond at Met86 in Grp94 (Met30 in Hsp90) based on a "coiled-coil" structure of the Trap1 NTD[20,21]. As discussed later, a recent cryoEM structure of cytosolic Hsp70/Hsp90 provides an alternative structural model for the Grp94 C′ state (see loading step in Fig. 1). In this work, we will refer to the Grp94 C′ state as a client loading conformation.

BiP also undergoes an ATP-driven conformational cycle[22]. When ATP is bound, BiP adopts a conformation in which the substrate-binding domain (SBD) is docked onto the NBD, opening a lid that allows clients to transiently bind the SBD[23]. ATP hydrolysis by BiP causes the SBD to undock, which in many, but not all cases[24], results in stable lid closure over the bound client. Grp94 can directly bind to BiP when the SBD is undocked and the lid is closed, but not when BiP is in the ATP conformation[11]. This mechanism enables BiP to deliver client proteins to Grp94. The Grp94 K467A mutation on the middle domain (MD) at interface I abolishes BiP binding by breaking a key salt-bridge with BiP (BiP$_{E243}$:Grp94$_{K467}$)[11]. The Grp94 K467A mutation interferes with the folding of a subset of Grp94 clients in-vivo[25], which demonstrates that client delivery from BiP to Grp94 is important for cellular protein folding.

A cryoEM structure of the cytosolic Hsp70/Hsp90 pair has provided new insights[10]. The structure shows Hsp90 in a "semi-closed" loading conformation where the NTDs are rotated similarly to the closed state. Hsp70 interacts with Hsp90 through two interfaces. Interface I (bridging the Hsp70 NBD and Hsp90 MD) had been previously established as evolutionarily conserved and critical for the BiP/Grp94 interaction. The key BiP:Grp94 salt-bridge (BiP$_{E243}$:Grp94$_{K467}$) is similarly formed at interface I for Hsp70:Hsp90 (Hsp70$_{E218}$:Hsp90$_{K419}$). The novel interface II (between the Hsp70 NBD and Hsp90 NTD, see loading step in Fig. 1) was proposed to stabilize the Hsp90 dimer arms in the semi-closed configuration, but this has not been experimentally tested. It is important to test whether structures and mechanisms proposed for the cytosol-specific Hsp70/Hsp90 system also apply to other family members, such as the

BiP/Grp94 system, because the extent of mechanistic conservation is not yet clear. For instance, a recent study proposed that Grp94 can act upstream of BiP when acting on thermally denatured luciferase[26], which is opposite to the canonical behavior for the cytosolic Hsp70/Hsp90 system.

Here we examine client loading from BiP to Grp94, and how this process is impacted by inhibitors. We discover that some inhibitors block the Grp94 client loading conformation and destabilize the BiP/client/Grp94 ternary complex, whereas others do not. We propose that select inhibitors block loading because they would sterically clash with the pocket in the loading conformation. Removing one such clashing residue restores Grp94's ability to adopt the loading conformation for an inhibitor that would otherwise block loading. Using a monomeric construct of proIGF2, a biological client protein of Grp94[27–29], we show that client loading and Grp94 closure are both critically dependent on BiP's contacts to Grp94 at interface II. The results show that the Hsp70/Hsp90 systems in the ER and cytosol have specific shared features in client loading and explain why a select group of inhibitors disrupt client loading from BiP to Grp94.

## Results
### A select group of inhibitors block the Grp94 loading state
A previously described bulk FRET assay shows that the BiP NBD increases Grp94 FRET efficiency by driving Grp94 into the high-FRET C′ state[20]. These experiments are performed with the BiP NBD because this domain is sufficient to drive Grp94 into the C′ state[20] and avoids the confounding factor associated with the oligomerization of full-length BiP[30]. Given that Hsp90 inhibitors clash with the closed conformation (Supplementary Fig. 1), we initially expected that all Hsp90 inhibitors would prevent BiP-induced closure of Grp94 to the C′ state. However, in the presence of BiP, Hsp90 inhibitors have variable effects on the Grp94 conformation as indicated by variable changes in bulk FRET efficiency (Supplementary Fig. 2A). Supplementary Fig. 2B ranks a collection of commercially available inhibitors by the degree to which they change the Grp94 conformation. HSP990 and XL888 show maximal suppression of BiP-induced Grp94 FRET changes while AUY922 shows no suppression. While the bulk FRET data show that Hsp90 inhibitors can have variable effects on the Grp94 conformation, bulk FRET measurements provide population averages and cannot specify which Grp94 conformations are adopted.

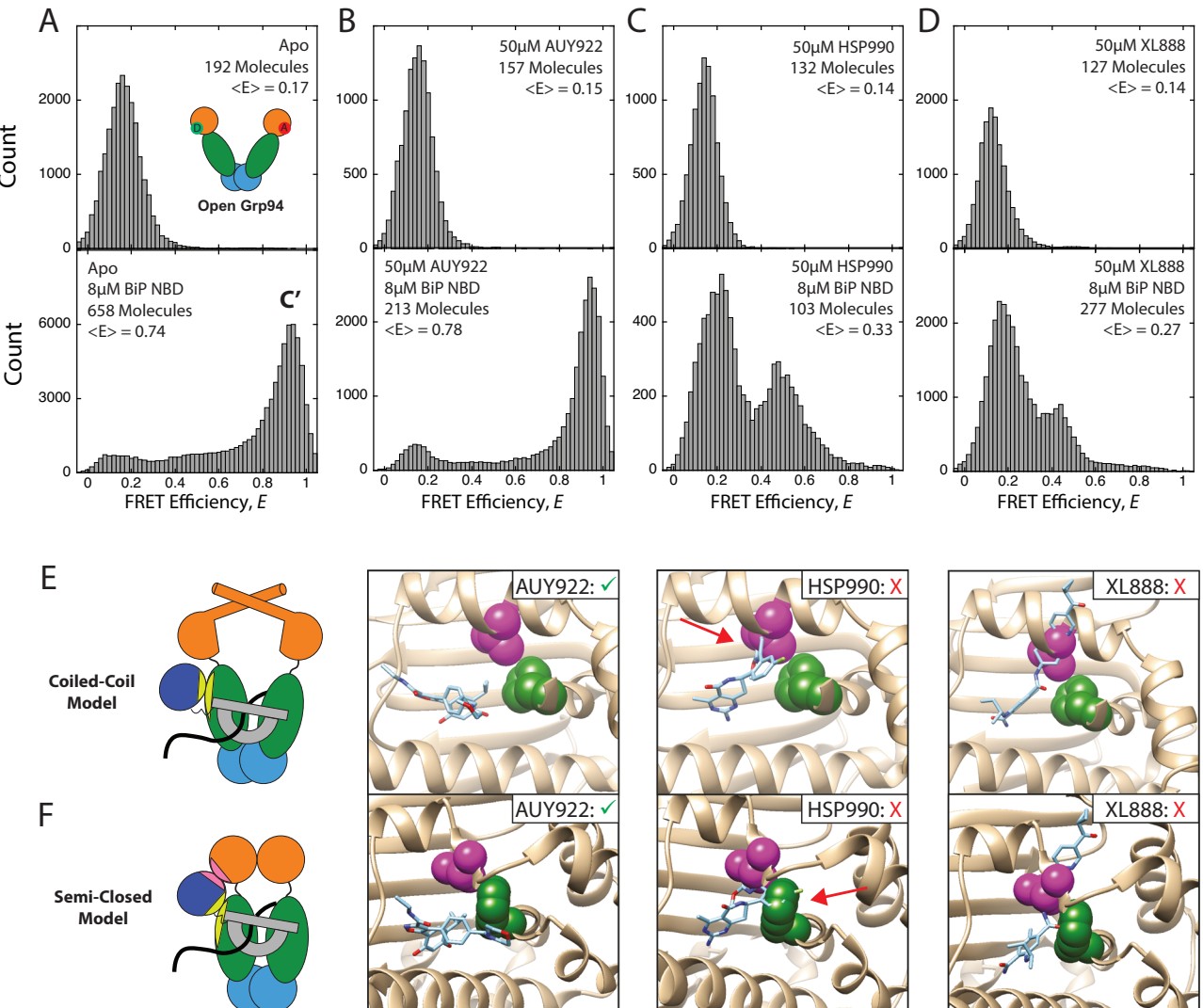

**Fig. 2 | Effect of inhibitors on Grp94 conformation. A** Grp94 smFRET efficiency histograms with and without 8 μM BiP NBD. Results are shown for measurements in the absence of any ligand (Apo, **A**), and in the presence of 50 μM AUY922 (**B**), 50 μM HSP990 (**C**), 50 μM XL888 (**D**). <E> is the average FRET efficiency for the entire histogram. See Methods for experimental details. **E** Superposition of AUY922 (PDB: 6LTI), HSP990 (PDB: 4U93), and XL888 (PDB: 4AWO) onto the coiled-coil NTD structure (PDB: 5F3K). **F** Superposition of AUY922, HSP990, and XL888 onto the semi-closed NTD structure (PDB: 7KW7). Inhibitors are shown in light blue. Residues corresponding to Hsp90α_{L107} and Hsp90α_{F138} are shown in magenta and green respectively. In (**A**) the red circle with (**A**) and green circle with D on open Grp94 represent acceptor and donor fluorophore labels at Grp94_{N91C}. Red arrows in (**E**, **F**) point to clashes made between HSP990 and specific residues. Source data are provided as a Source Data file.

AUY922, HSP990, and XL888 were selected for a more detailed smFRET analysis. Bulk measurements show that the inhibitors fully suppress Grp94 ATPase activity (Supplementary Fig. 3), which demonstrates that inhibitor binding is saturated at concentrations used in our smFRET experiments. In the absence of BiP, smFRET measurements show that none of the inhibitors influence the Grp94 conformation, which is maintained in the open state ($E \sim 0.15$, top panels in Fig. 2A–D). This result is consistent with previous studies of other Hsp90 family members in which inhibitors maintain Hsp90 in an open conformation[31]. In contrast, in the presence of BiP different outcomes are observed for each inhibitor. With AUY922, the Grp94 C' state ($E \sim 0.9$, lower panel Fig. 2B) remains highly populated, whereas with XL888 and HSP990, the C' state is minimally populated and Grp94 instead adopts a mixture of the open state (low FRET) and configurations with intermediate FRET ($E \sim 0.4$-$0.5$, lower panels Fig. 2C, D) that have not been observed in previous smFRET analysis of Grp94[20]. Example FRET, donor fluorescence, acceptor fluorescence, and direct acceptor excitation fluorescence traces are shown in Supplementary

Fig. 4. It is not clear whether these intermediate FRET efficiencies arise from a conformational intermediate in the normal process of Grp94 closure but too transient to detect, or whether the inhibitors have stabilized a novel off-pathway conformation.

A comparison of the average FRET efficiency measured by smFRET (< E > values in Fig. 2) and the bulk FRET show that these assays are reporting on the same trends in the conformation of Grp94 (Supplementary Fig. 5). Collectively, these results show that some inhibitors allow BiP to push Grp94 into the C' conformation, whereas other inhibitors block this conformational change. However, this data does not explain why only some inhibitors block the Grp94 C' conformational change, and how BiP stabilizes the Grp94 C' conformation in the first place.

## Steric clashes explain why select inhibitors block Grp94 loading conformation

Rather than focus on the Grp94 conformations with intermediate FRET efficiency that become populated from HSP990 and XL888, we wanted

to understand why Grp94 can populate the C′ state when bound to AUY922 but not when bound to HSP990 and XL888. We turned to two potential models of the Grp94 C′ state for insight: the Trap1 coiled-coil structure and the Hsp90α semi-closed structure (Fig. 1).

Previous work proposed that the Grp94 N-terminal α-helix in the C′ state has a configuration like the Trap1 coiled-coil structure[21]. This proposal came from results with a designed disulfide bond for Grp94_M86C that is predicted to form in the coiled-coil conformation[21]. Based on the smFRET measurements in Fig. 2, we would expect comparable disulfide bond formation in the absence of inhibitor and with the C′-compatible inhibitor AUY922, and expect decreased cross-linking for the C′-incompatible inhibitors HSP990 and XL888. We indeed observe these results (Supplementary Fig. 6). This shows that the Grp94_M86C crosslink continues to be a reliable test of the C′ state. However, these results do not address why the inhibitors have differential compatibility with the C′ state. To explore this question, we looked closer at the ATP-binding pockets in each structural model.

Both the coiled-coil structure and the semi-closed structure have no bound ligand and show notable conformational changes in the nucleotide binding pocket. For the semi-closed structure, a conserved Phe residue stands out (Hsp90α_F138, Trap1_F205, and Grp94_F199). A rotamer change around the $C_\alpha$-$C_\beta$ bond flips the Phe sidechain, creating a hydrophobic cluster with multiple lid residues (Supplementary Fig. 7A), including a conserved Leu (Hsp90α_L107, Trap1_L172, and Grp94_L163). Unlike the open state where the lid is highly dynamic[32,33], in the semi-closed conformation the lid is stable because it is part of the dimer interface (teal residues Supplementary Fig. 7B). For the Trap1 coiled-coil structure, no rotamer change is observed for Trap1_F205, but a different region of the lid is rearranged and projects the conserved Leu residue (Trap1_L172) into the nucleotide binding pocket. Therefore, steric clashes within the nucleotide pocket may explain the C′ incompatibility observed with XL888 and HSP990. This explanation can be tested by observing whether Grp94 can repopulate the C′ state by removing a candidate clashing residue.

Figure 2E, F shows a super-positioning of AUY922, HSP990, and XL888 onto the nucleotide binding pocket of the coiled-coil and semi-closed structures. Both structures are sterically compatible with AUY922 and clash with HSP990 and XL888, which is consistent with the smFRET observation that Grp94 can populate the C′ conformation when bound to AUY922 but not when bound to HSP990 and XL888 (Fig. 2B-D). For XL888, both the coiled-coil and semi-closed structures show extensive clashes with the peptide backbone of the lid, so a single mutation would not be expected to completely relieve the clashes. In

contrast, the residues that clash with HSP990 are more localized. For the coiled-coil structure HSP990 clashes with the Leu sidechain (arrow in Fig. 2E), whereas for the semi-closed structure HSP990 clashes with the Phe sidechain (arrow in Fig. 2F).

We constructed the Grp94_L163A and Grp94_F199A variants to see if either or both mutations allows Grp94 to repopulate the C′ state when HSP990 is bound to Grp94. AUY922, HSP990, and XL888 still suppress ATPase activity of the Grp94_L163A and Grp94_F199A variants (Supplementary Fig. 8), demonstrating that inhibitor binding remains saturated. The Grp94_L163A and Grp94_F199A variants have altered ATP turnover rates compared to wild-type, which is not surprising since the mutations are in the nucleotide binding pocket. We next fluorescently labeled Grp94_L163A and Grp94_F199A for bulk FRET measurements like those shown in Supplementary Fig. 2. The Grp94_L163A and Grp94_F199A FRET levels remain high with AUY922 and low with XL888 (Fig. 3A). For HSP990, the F199A mutation increases Grp94 FRET efficiency whereas L163A results in a modest decrease, consistent with a steric clash between HSP990 and F199 being a key contributor to why HSP990 is incompatible with the Grp94 loading state.

To ensure the increase in Grp94_F199A FRET with HSP990 is due to an accumulation of C′ state, we used Grp94_F199A with HSP990 in smFRET experiments in the presence and absence of 8 μM BiP NBD. Consistent with other experiments in the absence of BiP NBD, Grp94_F199A primarily populates the low FRET open state (Fig. 3B). A low population of higher FRET efficiency values is also observed, perhaps due to mild destabilization of Grp94 by the F199A. Notably in the presence of BiP and HSP990, the Grp94_F199A peak FRET efficiency E ~ 0.9 confirms that the C′ state is populated (Fig. 3C).

We also performed smFRET experiments with Grp94_F199A and Grp94_L163A in the presence of the different inhibitors. Grp94_L163A in the absence and presence of BiP populates the open state and C′ state respectively, as expected (Supplementary Fig. 9A). Likewise, Grp94_F199A and Grp94_L163A in the presence of BiP and AUY922 populate the C′ state (Supplementary Fig. 9B). However, the smFRET results with XL888 or HSP990 are more complex (Supplementary Fig. 9C and D). For both Grp94_F199A and Grp94_L163A in the presence of BiP and XL888, minimal open state is populated whereas a broad intermediate FRET peak is observed with variable mean FRET efficiencies (Supplementary Fig. 9C and D). The broad range of FRET efficiencies may represent multiple conformational states perhaps due to destabilizing effects of F199A and L163A. Similar results were observed for Grp94_L163A in the presence of BiP and HSP990 (Supplementary Fig. 9D). The conformational equilibrium between open and intermediate FRET states is

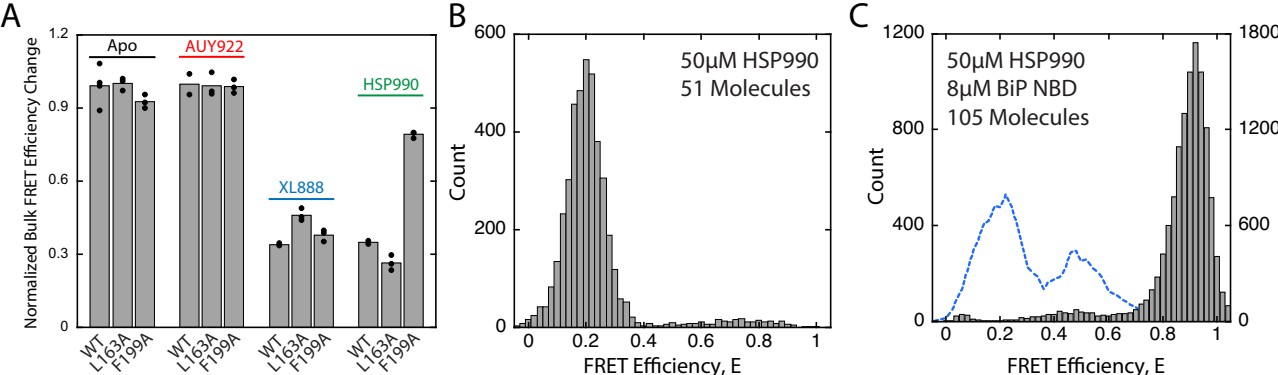

**Fig. 3 | Role of Grp94_F199A for Grp94 loading conformation with HSP990.**
**A** Normalized Grp94 bulk FRET efficiency changes at 8 μM BiP NBD for wild-type Grp94, Grp94_L163A, and Grp94_F199A with no inhibitor or 50 μM inhibitor. Data points are from independent replicate measurements. See Supplementary Fig. 2 and 10 for complete BiP concentration series. **B** smFRET efficiency histograms for Grp94_F199A

in the presence of 50 μM HSP990. **C** Overlay of smFRET histograms for wild-type Grp94 (blue dashed represents outline of histogram from the lower panel of Fig. 2C) and Grp94_F199A (gray bars) in the presence of 50 μM HSP990 and 8 μM BiP NBD. Count values on left and right x-axes are for wild-type Grp94 and Grp94_F199A, respectively. Source data are provided as a Source Data file.

thus sensitive to the F199A and L163A mutations, with the open state being readily destabilized by mutations in the ATP binding pocket. Although a small C′ population is observed for Grp94$_{L163A}$ bound to XL888, no other combination of Grp94 mutant and inhibitor results in a wholesale shift to the C′ population as was seen for Grp94$_{F199A}$ in the presence of BiP and HSP990 (Fig. 3C).

Collectively these results suggest that sterics explain why some inhibitors allow BiP to push Grp94 into the loading state whereas other inhibitors block this change. Specifically, XL888 and HSP990 would clash with the nucleotide binding pocket in the loading state. In the case of HSP990, we conclude that the clash is primarily with F199.

## BiP drives Grp94 conformational changes via contacts at interface II

We next asked how BiP pushes Grp94 into the loading conformation. For the cytosolic Hsp90 loading structure, it has been proposed that the contact between Hsp70 and Hsp90 at interface II stabilizes the semi-closed state (Fig. 1). We constructed two BiP variants that would be expected to disrupt interface II (BiP$_{D187A}$ and BiP$_{II91A}$, Fig. 4A), and find that these mutations abolish BiP-induced conformational changes of Grp94. This can be seen with the bulk FRET assay in which wild-type BiP NBD increases Grp94 FRET efficiency, whereas no change is observed for BiP$_{D187A}$ and BiP$_{II91A}$ NBD (Fig. 4B). Furthermore, for BiP$_{D187A}$ and BiP$_{II91A}$, no differences are observed between AUY922, HSP990, and XL888 on Grp94 conformation (Supplementary Fig. 11). We conclude that the higher FRET states of Grp94 that are induced by

BiP, both with and without inhibitors (*i.e.* the loading conformation and intermediate FRET state from Fig. 2), are a result of contacts at interface II.

Previous work demonstrated that BiP accelerates ATP-dependent closure of Grp94, resulting in accelerated ATP turnover[20]. To determine the role of interface II on BiP's stimulation of Grp94 ATP turnover, we utilized a BiP NBD construct that is hydrolytically inactive and unable to bind ATP (termed NBD*, a variant combining the T229G and G227D mutations[34]) so the only ATPase activity comes from Grp94. Figure 4C shows that NBD* enhances Grp94 ATPase activity, similar to previous results using an NBD construct that can bind ATP[11]. Importantly, BiP$_{D187A}$ and BiP$_{II91A}$ NBD greatly diminish the enhanced activity of Grp94. We next measured the rate of ATP-dependent closure of Grp94 with a previously established bulk FRET assay[20]. NBD* accelerates Grp94 closure, whereas no acceleration is observed for BiP$_{D187A}$ and BiP$_{II91A}$ NBD (Fig. 4D). Because NBD* cannot bind ATP we can exclude a model in which the activation of Grp94 by BiP depends on the identity of the nucleotide bound within the NBD.

Interface II is not only responsible for BiP-induced conformational changes to Grp94, but this region also makes a substantial contribution to the binding affinity between BiP and Grp94. A fluorescence polarization (FP) assay in Fig. 4E shows that the Grp94 affinity to the BiP NBD is reduced ~6-fold by interface II mutations. The affinity of Grp94 to BiP$_{D187A}$ and BiP$_{II91A}$ NBD is similar to the affinity of the Grp94 MD to the wild-type BiP NBD. This comparison shows that BiP$_{D187A}$ and BiP$_{II91A}$ remove the entire affinity contribution from interface II, leav-

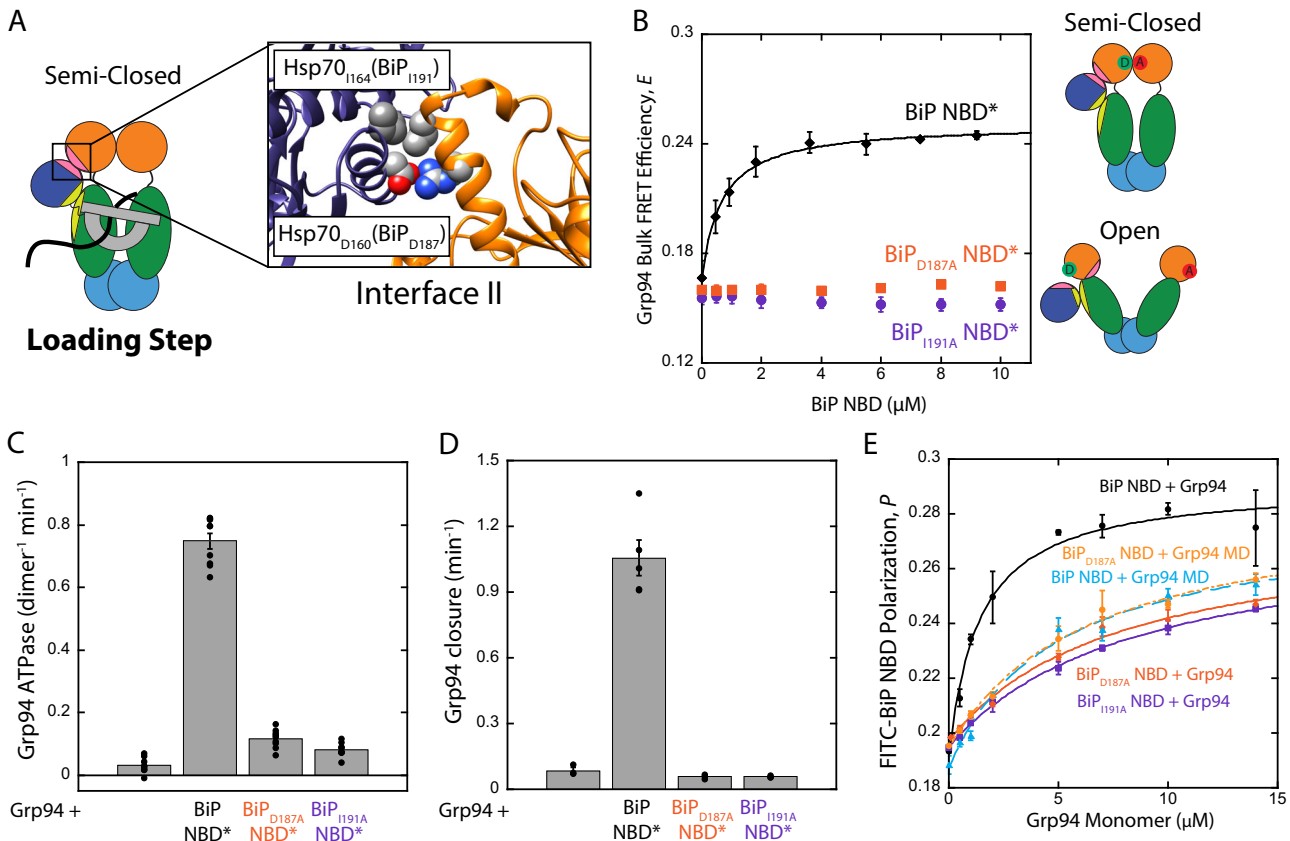

**Fig. 4 | Role of interface II connecting BiP and Grp94. A** Predicted sites of BiP$_{II91}$ (Hsp70$_{I164}$) and BiP$_{D187}$ (Hsp70$_{D160}$) at interface II (PDB: 7KW7). **B** Grp94 bulk FRET efficiency under apo conditions. The solid line is fit to a single site binding equation (K$_{d,app}$: 0.67 ± 0.07 μM). **C** Grp94 ATPase rates in the absence and presence of 2 μM BiP NBD constructs. **D** Grp94 ATP-dependent closure rates in the absence and presence of 1 μM BiP NBD constructs. **E** FP BiP binding assay of either Grp94 or Grp94 MD to different FITC-labeled NBD constructs under ADP conditions. Lines

are fit to a single-site binding equation. K$_d$ values from the fitting: 1.40 ± 0.02 μM (BiP NBD + Grp94), 5.8 ± 1.2 μM (BiP NBD + Grp94 MD), 8.2 ± 1.0 μM (BiP$_{D187A}$ NBD + Grp94), 9.0 ± 1.7 μM (BiP$_{II91A}$ NBD + Grp94), 6.5 ± 0.7 μM (BiP$_{D187A}$ NBD + Grp94 MD). Error bars are the SEM from independent replicate measurements ((**B**) $n = 3$, **C**) $n = 9$, (**D**) $n = 3$ except for BiP NBD* where $n = 5$, (**E**) $n = 3$ except Grp94 MD with FITC-NBD and FITC-D187A NBD where $n = 6$ and $n = 5$, respectively. Source data are provided as a Source Data file.

ing only the interface I contribution at the Grp94 MD. $BiP_{D187A}$ does not impact MD binding, which excludes the possibility that the interface II mutations have interfered with BiP binding via changes to interface I on the Grp94 MD.

The above results show that interface II is critical to how BiP/Grp94 work together. First, interface II provides the driving force by which BiP stabilizes the loading conformation of Grp94 (Fig. 4B). Second, by stabilizing the loading conformation, interface II is the driving force behind BiP acceleration of Grp94 ATP-dependent closure[20] (Fig. 4D). Because interface II makes a major contribution to the BiP/Grp94 affinity (Fig. 4E) we can now anticipate that loading-incompatible inhibitors (XL888 and HSP990) should disrupt a BiP/client/Grp94 ternary complex to a greater extent than AUY922. We confirm this prediction later with the proIGF2 client.

### A monomeric proIGF2 client construct for client loading analysis

We next asked about how client loading occurs for proIGF2, the pro-protein of insulin-like growth factor 2 (IGF2). ProIGF2 is a well-established Grp94 client due to the essential biological role Grp94 plays in IGF2 folding and secretion of the mature hormone[27–29]. Previous in-vitro studies demonstrated that the E-peptide region of proIGF2 promotes oligomerization[35], which is a common characteristic of peptide hormones that has been proposed to enable efficient sorting in the secretory pathway and storage in secretory granules[36]. The main BiP binding site on proIGF2 ("site 1") has been identified on the N-terminal end of the E-peptide[37]. Because proIGF2 oligomerization complicates experiments needed to analyze client loading on Grp94, we set out to establish a monomeric proIGF2 construct that includes the primary BiP binding site and is long enough to reach the Grp94 binding cleft while simultaneously bound to BiP.

We designed a proIGF2 truncation past the site 1 region on the E-peptide with these goals in mind ($proIGF2_{25-120}$, Fig. 5A). Five of the six native Cys residues on $proIGF2_{25-120}$ are mutated to Ser, to prevent oxidative folding of the mature IGF2 region (mIGF2, Fig. 5A). The remaining Cys allows fluorophore labeling for FP measurements. Under ADP conditions, BiP binds $proIGF2_{25-120}$ with a $K_D$ ($2.5 \pm 0.4\,\mu M$) that is comparable to the isolated site 1 peptide ($2.5 \pm 0.7\,\mu M$) and an "extended site 1" construct containing proIGF2 residues 92–120 ($1.9 \pm 0.5\,\mu M$) (Supplementary Fig. 12A-C). Interestingly, we observe a trend under ATP conditions in which BiP binds the progressively longer client constructs with progressively higher affinity (Supplementary Fig. 12D).

We performed analytical size exclusion chromatography (SEC) to examine the oligomeric state of $proIGF2_{25-120}$. The retention profile of $proIGF2_{25-120}$ shows a single peak with a retention time consistent with a monomer, according to protein standards (Fig. 5B). ProIGF2 and E-peptide cannot be analyzed by SEC because they form large oligomers that interact with the resin, but dynamic light scattering (DLS) measurements can quantify their mean hydrodynamic radius ($R_H$). At a concentration of $1\,\mu M$, proIGF2 oligomers have an $R_H$ value of $1120 \pm 80$ nm, E-peptide oligomers have an $R_H$ value of $330 \pm 50$ nm, whereas $proIGF2_{25-120}$ produces so little light scattering that an $R_H$ value could not be determined reliably. Together, the SEC, DLS, and FP therefore indicate that $proIGF2_{25-120}$ is monomeric and binds BiP.

Under ADP conditions, $BiP:proIGF2_{25-120}$ complex formation is evident by SEC coelution (Fig. 5C, upper panel). In these experiments, we used FITC-labeled $proIGF2_{25-120}$, which enables the client retention to be tracked by absorption at 495 nm while simultaneously monitoring BiP by the absorption at 280 nm (the BiP extinction coefficient at 280 nm is 28,880 $M^{-1}\,cm^{-1}$ is greater than the $proIGF2_{25-120}$ extinction coefficient of 12,950 $M^{-1}\,cm^{-1}$). Under ATP conditions, two BiP retention peaks are observed (Fig. 5C, lower panel). The first retention peak contains $proIGF2_{25-120}$ and matches the retention time observed under ADP conditions (8.8 min). The second retention peak does not

contain proIGF25-120 and matches the retention time of BiP alone at 9.2 min.

While the SEC results are consistent with BiP ATP hydrolysis trapping $proIGF2_{25-120}$ by fully closing the lid, previous findings with a different large client protein ($C_H1$) indicated no such lid closure of BiP[24]. To determine whether the BiP lid closes with $proIGF2_{25-120}$, we utilized a previously established bulk FRET assay in which BiP has low FRET when the lid is open and high FRET when the lid is closed[24]. We observe a BiP FRET efficiency with $proIGF2_{25-120}$ that is comparable to the ADP-state of BiP in which the lid is fully closed (Fig. 5D). We also find that $proIGF2_{25-120}$ accelerates BiP lid closure. This acceleration is due to the site 1 region, since its removal results in no lid closure acceleration (see BiP + mIGF2 in Fig. 5D). Overall, we conclude that $proIGF2_{25-120}$ is a well-behaved monomeric client construct that can be stably trapped by BiP in the lid closed state. This establishes $proIGF2_{25-120}$ as well-suited for examining client loading from BiP to Grp94 and how loading is impacted by inhibitors.

### Effect of inhibitors and interface II on client loading from BiP to Grp94

Unlike the stable binding between BiP and $proIGF2_{25-120}$, FP measurements show that Grp94 by itself has minimal client binding affinity (Fig. 6A). However, substantially larger FP values are observed when BiP and Grp94 are together, demonstrating that $BiP/proIGF2_{25-120}/Grp94$ form a large, slowly-tumbling ternary complex. The apparent affinity of $proIGF2_{25-120}$ to BiP/Grp94 is comparable to BiP alone (compare red and green curves in Fig. 6A), but this should be interpreted cautiously because the FP signal likely has contributions from $proIGF2_{25-120}$ binding to both BiP and BiP/Grp94 over the experimental concentration range. FITC-labeled mIGF2 exhibits a minimal increase of FP from BiP/Grp94 (Fig. 6B), indicating that BiP's binding to the site 1 region is necessary for forming the $BiP/proIGF2_{25-120}/Grp94$ ternary complex.

The increase in FP signal from Grp94 binding to $BiP:FITC-proIGF2_{25-120}$ ($\Delta P$, the arrow shown in Fig. 6A) is a measure of BiP/client/Grp94 ternary complex formation. Figure 6C shows that the interface I mutation $Grp94_{K467A}$ results in a total knockout of the ternary complex due to the inability of BiP to bind Grp94, as expected. The interface II mutations ($BiP_{D187A}$ and $BiP_{I191A}$) knockdown the ternary complex to a level only slightly above $Grp94_{K467A}$, which demonstrates that interface II plays a critical role in the stability of the client loading complex.

Loading-incompatible inhibitors XL888 and HSP990 disrupt the BiP/client/Grp94 ternary complex whereas the loading-compatible inhibitor AUY922 modestly enhances ternary complex stability (red arrows in Fig. 6C). We also tested the combined impact of interface II mutations and inhibitors, and observed only minor differences between AUY922, XL888, and HSP990 in this context (Supplementary Fig. 13). Interestingly, the XL888 and HSP990 $\Delta P$-values are higher than for the interface II mutations (Fig. 6C compare bars 3 and 4 with bars 6 and 7), which means that XL888 and HSP990 have not disrupted ternary complex formation to a level expected for a complete loss of interface II. This suggests room for improvement in designing Hsp90 inhibitors that destabilize client loading to an even greater extent than either XL888 or HSP990.

### Discussion

Here we find that functional differences between Hsp90 inhibitors become apparent when Grp94 is working with BiP because Hsp90 inhibitors have variable compatibility with the Grp94 loading state (Fig. 2). It should not be surprising that inhibitors have differing compatibilities with the loading state, because its structure and existence were not considered in inhibitor design. We propose that inhibitor incompatibility with the Grp94 loading state can be explained by steric clashes. Based on the cytosolic semi-closed Hsp90 structure we

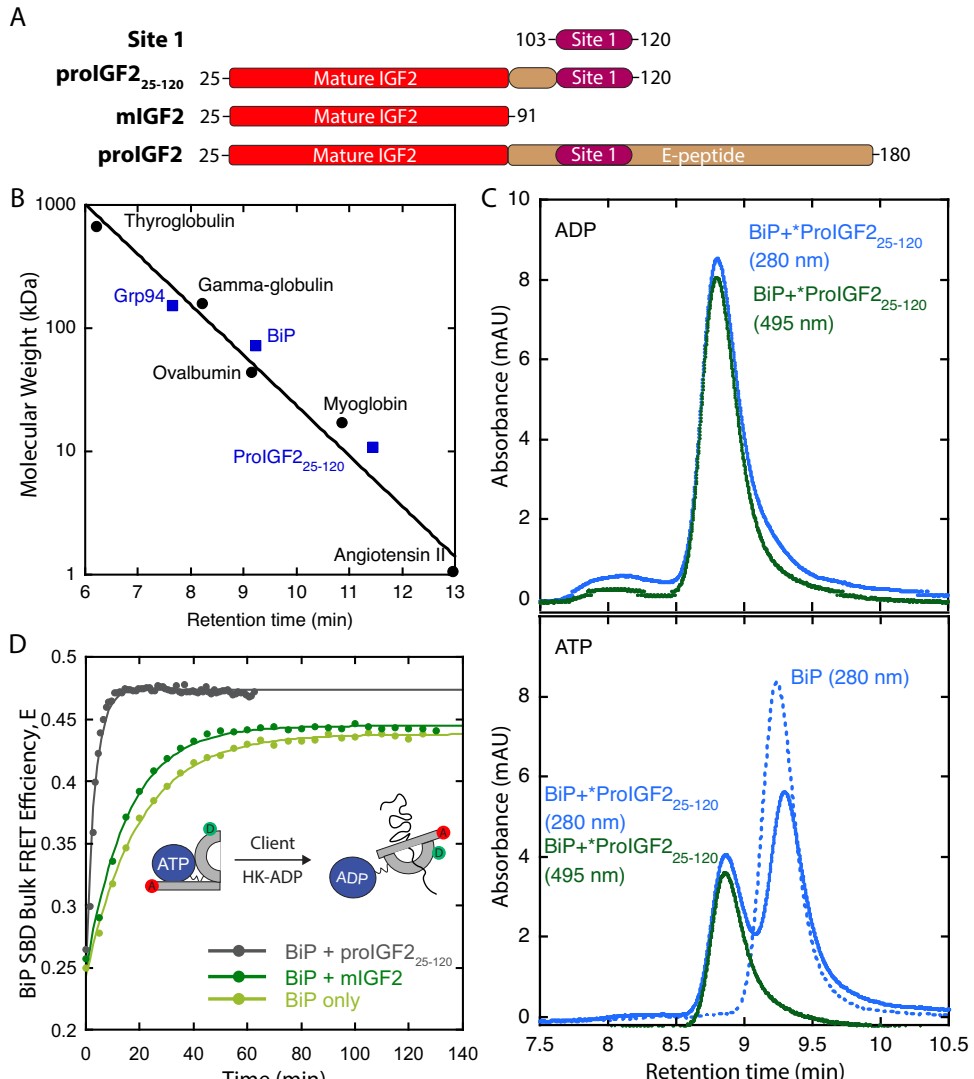

**Fig. 5 | A monomeric proIGF2 client construct. A** ProIGF2 client fragments: E-peptide (tan), mIGF2 (red), and site 1 (purple). **B** SEC retention time versus molecular weight of standard proteins (black circles). Positions of a proIGF2$_{25\text{-}120}$ monomer, BiP, and Grp94 dimer are indicated in blue squares. **C** Upper panel shows retention profile of BiP and FITC-labeled proIGF2$_{25\text{-}120}$ (*ProIGF2) under ADP conditions as measured by absorption at 280 nm (blue) and 495 nm (green). Lower panel shows the same experiment under ATP conditions, as well as the retention of BiP in the absence of client (dashed blue line). **D** BiP SBD bulk FRET assay measuring

SBD lid closure kinetics with and without client. Solid lines are a fit to a single exponential (closure rate with 5 μM proIGF2$_{25\text{-}120}$: $0.30 \pm 0.01$ min$^{-1}$; 5 μM mIGF2: $0.06 \pm 0.01$ min$^{-1}$; without client: $0.05 \pm 0.01$ min$^{-1}$). Uncertainty on closure rates is the fitting error. Closure is initiated by first equilibrating BiP with 0.1 mM ATP and flushing in 1 mM of ADP with hexokinase and glucose (HK-ADP, see Methods) and client. Cartoon shows schematic of BiP conformations and fluorophore positions. Source data are provided as a Source Data file.

identify F199 as one such clashing residue and show that F199A restores Grp94's ability to adopt the loading conformation for the otherwise loading-incompatible inhibitor HSP990 (Fig. 3). These results open the possibility of rational drug design based off the semi-closed structure, which may enable greater control over the biological outcome of Hsp90 inhibitors.

It will be interesting to see whether differences between AUY922, HSP990, and XL888 are also observed for client loading for the cytosol-specific Hsp70/Hsp90 system. If so, inhibitor compatibility with the loading state would be relevant for clients such as steroid hormone receptors and p53, which can be loaded from Hsp70 to Hsp90[38–42]. In contrast, it is not clear whether the loading compatibility of inhibitors would be relevant for kinase clients, because these can bypass Hsp70 and be loaded directly from co-chaperones such as cdc37[43].

To evaluate the effect of inhibitors on client loading we designed proIGF2$_{25\text{-}120}$ (Fig. 5), which is unfolded, monomeric, includes the

primary BiP binding site, and is long enough to reach the Grp94 binding cleft while simultaneously bound to BiP. Using this construct, we find that the loading-incompatible inhibitors (HSP990 and XL888) disrupt the BiP/client/Grp94 ternary complex whereas the loading compatible inhibitor AUY922 results in a modest stabilization (Fig. 6C). The ability of inhibitors to cause clients to be retained on BiP versus in complex with Grp94 is important because BiP and Grp94 interact with different pro-folding and pro-degradation components of the ER quality control system[22], which suggests loading compatible inhibitors may have different outcomes than loading incompatible inhibitors. XL888 and HSP990 do not disrupt ternary complex formation to the extent expected for a total loss of interface II. One possible explanation may be related to the smFRET results in Fig. 2, which show that XL888 and HSP990 prevent BiP from driving Grp94 into the loading conformation but also cause Grp94 to adopt a new conformation with intermediate FRET efficiency. It is possible that this conformation of Grp94 confers a modest stabilization of the BiP/client/Grp94 ternary

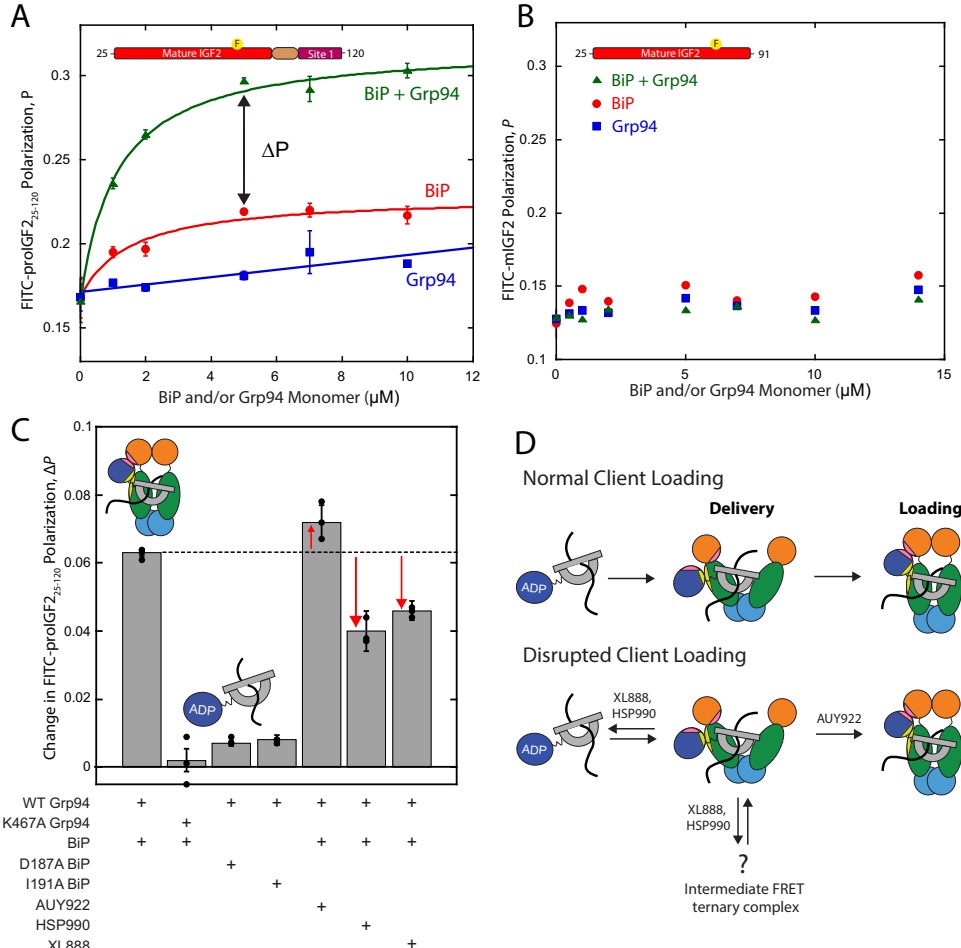

**Fig. 6 | Influence of inhibitors on BiP-Grp94-client ternary complex. A** FP of FITC-labeled proIGF2$_{25-120}$ with varying concentrations of BiP (red), Grp94 (blue), and BiP and Grp94 (green) under ADP conditions. Solid lines for BiP ($K_D = 1.5 \pm 0.7$ μM) and BiP/Grp94 ($K_{D,app} = 1.1 \pm 0.2$ μM) are fits to a single-site binding equation. Grp94 only data is a linear fit. Double-headed arrow represents increase in polarization due to Grp94 association. Yellow circle with F represents labeling site of FITC on proIGF2$_{25-120}$. **B** FP of FITC-labeled mIGF2 with varying concentrations of BiP (red), Grp94 (blue), and BiP and Grp94 (green) under ADP conditions. **C** Change in FP (Δ$P$) of FITC-labeled proIGF2$_{25-120}$ with 5 μM BiP, 5 μM Grp94 monomer, and ADP, in the presence and absence of inhibitors, and various BiP and Grp94 mutants. **D**. Schematic of inhibitor structural effects on ternary complex formation of BiP, Grp94, and proIGF2 client. Error bars are SEM for three independent replicate measurements. Source data are provided as a Source Data file.

complex as compared to the complete removal of interface II (Fig. 6D). More work is needed to address this possibility.

Our findings with interface II (Fig. 4) show specific structural and mechanistic conservation in client loading shared between the Hsp90 systems in the ER and cytosol. Interface II is responsible for driving Grp94 into the loading conformation, plays a key role in stabilizing the BiP/client/Grp94 loading complex (Fig. 6C) and provides a structural explanation for how BiP acts as a closure-accelerating cochaperone for Grp94[20]. The BiP mutations (BiP$_{D187A}$ and BiP$_{I191A}$) that disrupt interface II will be useful tools for future studies dissecting the process of client transfer from BiP to Grp94.

Collectively, our results indicate that neither the semi-closed structure nor the coiled-coil structure of Hsp90 are completely representative of the Grp94 loading state. The semi-closed structure accurately predicts the key role of interface II as well as the clash between F199 and HSP990. In contrast, the M86C crosslink is not consistent with the semi-closed structure because Hsp90$_{M30}$ (corresponding to Grp94$_{M86}$) is buried in the semi-closed structure (Supplementary Fig. 14). But if the N-terminal α-helix were to detach from the NTD, which is a necessary step for complete arm closure, then a disulfide bond would be expected at this position for the semi-closed conformation (Supplementary Fig. 14 and Supplementary Table 1).

Meanwhile, the crosslinking measurements with inhibitors, based off of the disulfide bond predicted to form in coiled-coil structure, show the expected results based off of C′ incompatible and compatible inhibitors (Supplementary Fig. 6). Of note, a recent report has concluded that the BiP/Grp94 complex state more closely resembles the semi-closed conformation than the coiled-coil state[25]. More work is needed to determine the structure associated with the Grp94 C′ FRET state, and whether this FRET state can be described by a single conformation. Alternatively, it is possible that the semi-closed and coiled-coiled conformations of Grp94 are both populated, perhaps sequentially, in the process of closure.

## Methods

### Protein Expression

All proteins were expressed in *E. coli* BL21 cells. Mouse BiP (residues 27-655) and Grp94 (residues 72-765) with the charged-linker removed (Δ287-328) were first purified by Ni-NTA affinity chromatography. After cleavage of the N-terminal 6xHis-tag, proteins were further purified by anion exchange chromatography and size exclusion chromatography. Prior to flash freezing, proteins were stored in 25 mM Tris pH 7.5, 50 mM KCl, 1 mM 2-Mercaptoethanol (BME), and 5% glycerol. Fluorophore labeling of Grp94 N91C with Alexa Fluor 555 (donor) and

Alexa Fluor 647 (acceptor) was performed as described in previous work[19]. Grp94 N91C was fluorescently labeled for 3 h at room temperature. Grp94 was labeled at a monomer concentration of 25–30 µM with a 5-fold excess of dye. After labeling, a 4-fold molar excess of BME over dye was added to quench the reaction. Labeled Grp94 was separated from free dye via gel filtration. The fluorophore labeling efficiencies for all proteins, 70%–80%, were determined by absorption measurements.

For proIGF2$_{25-120}$ and mIGF2, all cysteines were mutated to serines except Cys70. ProIGF2$_{25-120}$ and mIGF2 were expressed in *E. coli* BL21 cells and cells were grown in LB at 37 °C. Protein expression was induced with 0.1 mM IPTG at an OD$_{600}$ between 0.6 and 0.8 and cells expressed overnight at 30 °C. ProIGF2$_{25-120}$ and mIGF2 were purified from inclusion bodies. Inclusion bodies were lysed by sonication in 100 mM Tris and 50 mM KCl. Pellet was washed with 4 M urea, 50 mM KCl, and 1% Triton-X until supernatant was visibly clear. Pellet was then washed with 100 mM Tris pH 7.5, 50 mM KCl, and 2 M NaCl to remove DNA. Insoluble protein was denatured in 8 M urea, 25 mM Tris pH 7.5, and TCEP. Protein was purified by cation exchange chromatography. Proteins were labeled with FITC-maleimide, with excess dye being removed by buffer exchange into 8 M urea and 25 mM Tris pH 7.5.

The proIGF2$_{92-120}$ fragment (Ext. Site 1) and site 1 used in binding experiments in Supplementary Fig. 12 have been described previously[37]. Ext. Site 1 was purified from inclusion bodies. Ext. Site 1 contained an N-terminal 6-histidine tag and cysteine mutation at Ser95 for FITC labeling. Briefly, inclusion bodies were washed and insoluble protein was denatured in an 8 M urea, 25 mM Tris buffer containing reducing agent TCEP. Ext. Site 1 was purified by ion-exchange chromatography and/or Ni-NTA affinity chromatography in denaturing conditions. Proteins used in FP assays were labeled with FITC-maleimide. Ext. Site 1 was stored denatured in buffer containing 8 M urea. Site 1 N-terminally labeled with FITC via an amino hexanoic acid linker was synthesized by Alan Scientific.

## ATPase measurements

Grp94 ATPase was measured by depletion of NADH via an enzyme-linked assay on a plate reader (BioTek). NADH depletion was monitored at an absorbance of 340 nm. Backgrounds were collected for 30 min to 1 h prior to the addition of Grp94. For ATPase measurements, 2 µM Grp94 dimer was assayed in 25 mM Tris pH 7.5, 50 mM KCl, 1 mM MgCl$_2$, 1 mg/mL BSA, 0.1 µM lactate dehydrogenase, 0.1 µM pyruvate kinase, 0.4 mM PEP, and 0.4 mM NADH at 30 °C. For samples with inhibitor, 50 µM was used. If full inhibition of Grp94 ATPase could not be achieved with 1 mM ATP, 50 µM ATP was used. All other samples used 1 mM ATP. The ATPase rate is reported per dimer of Grp94.

## Fluorescence Polarization (FP)

The method of FITC-labeling D27C BiP was performed as described in previous work[11] with a 3-fold excess of FITC under 25 mM Tris, pH 7.5, 50 mM KCl, and 0.25 mM TCEP at room temperature for 1 h. Labeling was quenched by adding 5 mM BME. Labeled protein was separated from free dye via gel filtration. Protein was stored in 25 mM Tris, pH 7.5, 50 mM KCl, 5% glycerol, and 1 mM BME. FP assays with labeled proIGF2$_{25-120}$ or mIGF2 used the S70C variant that was labeled with FITC similarly to BiP NBD. FP measurements were performed with 50 nM FITC-labeled BiP NBD or 50 nM FITC-labeled proIGF2$_{25-120}$ on a Fluoromax-4 spectrofluorometer (Horiba Scientific). Fluorometer setup had an excitation wavelength of 493 nm and an emission wavelength of 518 nm with 6 nm slit widths, and an integration time of 1 s. For FP experiments in Fig. 6 and Supplementary Fig. 12, the buffer conditions consist of 60 mM HEPES pH 7.0, 50 mM KCl, 5 mM MgCl$_2$, 5% or 0% DMSO v/v, 1 mM DTT, 0.75 mg/mL BSA, and 1 mM ADP at 30 °C. For BiP FP experiments, samples were incubated for 10 min prior to the addition of Grp94. For FP experiments in Fig. 4, the buffer condition was 25 mM Tris, pH 7.5, 50 mM KCl, 1 mM ADP, and 1 mM

DTT. In experiments with ADP, the ADP was treated with hexokinase for 1 h at 37 °C to remove any ATP contamination. BiP was incubated at 30 °C for 30 min prior to the experiments for deoligomerization. In experiments with FITC labeled proIGF2$_{25-120}$, proIGF2$_{25-120}$ and BiP were incubated at 30 °C for 30 min prior to the experiments to ensure client was bound to BiP.

For all BiP and mIGF2 FP experiments, the 30 min time point was used. For proIGF2$_{25-120}$ FP experiments, the 30 min time point was used if polarization values are constant over time. For proIGF2$_{25-120}$ FP experiments with observable kinetics, the data is fit to an exponential rise equation:

$$y = a + b(1 - e^{-c*t}) \tag{1}$$

Where $a$ is the starting polarization value, $b$ is the amplitude, $c$ is the rate, and $t$ is time. The plateau value of the FP is the reported value. $K_D$ values were calculated using the single-site binding equation:

$$P = \frac{a[x]}{K_D + [x]} + c \tag{2}$$

Where $P$ is the fluorescence polarization, $a$ is the polarization amplitude, $c$ is the polarization value in the absence of Grp94, and $x$ is the concentration of Grp94. All Grp94 titrations to determine $K_D$ values were from multiple separate experiments, each with a different concentration of Grp94.

## Single molecule FRET

The procedure for selectively biotinylating Grp94 with a C-terminal SNAP tag using a benzyl-guanine derivative was performed as previously described[20]. Acceptor-labeled SNAP-Grp94 (N91C) and donor-labeled Grp94 (N91C) were monomer exchanged at 30 °C for 2 h in a buffer consisting of 50 mM HEPES pH 8.0, 50 mM KCl, 0.6 mM MgCl$_2$, 2 mM BME and 0.5 mg/ml BSA. Preparation of the glass slides and coverslips has been described previously[19]. Grp94 was introduced into the chamber after dilution to 1 nM with an oxygen scavenging system (0.4% glucose, 1.5 units/µl catalase, 0.04 units/µl glucose oxidase) and triplet-state quencher cocktail (2 mM propyl galate, 4 mM 4-nitrobenzyl alcohol, and 4 mM Trolox). For experiments with ATP and inhibitors, the Grp94 sample was preincubated with the ligands. In relevant experiments, BiP NDB was then introduced into the chamber. Single molecule TIRF imaging was performed on a custom microscope as previously described[44].

To enable subtraction of background fluorescence for the donor and acceptor signals FRET analysis was only performed on molecules prior to a recorded donor photobleaching event. FRET efficiency values were only calculated for time points where neither donor nor acceptor had undergone photobleaching. Donor fluorophore excitation was at 532 nm with a laser power of 650 µW. All experiments included alternating excitation with an acceptor excitation at 633 nm with a laser power of 150 µW. The smFRET efficiency $E$ is calculated as

$$E = (\text{acceptor emission})/(\gamma^*(\text{donor emission}) + (\text{acceptor emission})) \tag{3}$$

where the $\gamma$ value is 1.75. This $\gamma$ value was determined through the same methods described previously[20].

## Grp94 Bulk FRET

Experiments were started by mixing 125 nM of donor- and acceptor-labeled Grp94 and incubating the mixture for at least 1 h at room temperature to allow for monomer exchange. For all experiments except for that in Fig. 4, the experimental conditions were 37 °C, 50 mM KCl, 25 mM HEPES pH 8.0, 600 µM MgCl$_2$, 1 mM BME, 0.75 mg/mL BSA, and 2% DMSO v/v. For Fig. 4B, the experimental

conditions were 30 °C, 25 mM Tris, pH 7.5, 50 mM KCl, 1 mM BME, and 1 mg/mL BSA. For Fig. 4D, the experimental conditions were the same as in Fig. 4B except that the experiments were at room temperature. All non-kinetic data was collected with a FluoroMax-4 spectrofluorometer. Data in Fig. 4D was collected with a Duetta spectrofluorometer (Horiba Scientific). For samples with inhibitor, 50 μM was used unless specified otherwise, and samples with ADP or ATP used 1 mM with a matching concentration of $MgCl_2$. The donor was excited at 532 nm and emission was measured at 565 nm and 670 nm for the donor and acceptor. Slit widths were set to 1.5 nm for excitation and 4.5 nm for emission. FRET efficiency was calculated from the emission as acceptor/(acceptor + donor). In Fig. 3A and Supplementary Fig. 10, FRET efficiency changes were calculated by subtracting the lowest FRET efficiency value of a dataset from all data points (dataset includes Apo, AUY922, ADP, Hsp990, and XL888 FRET measurements for a singular Grp94 variant). Normalized FRET efficiency changes were calculated by dividing all data points by the maximal FRET efficiency change value in a dataset.

### Analytical Size Exclusion Chromatography
The retention profile of FITC-proIGF2$_{25-120}$ was determined by diluting the client to 5 μM in HPLC Buffer (25 mM Tris pH 7.5, 50 mM KCl, 1 mM $MgCl_2$). AdvanceBio SEC 300 Å Protein Standards (Agilent Technologies) were used for molecular weight calculations. For samples including BiP, BiP was diluted to 4 μM in HPLC Buffer with either 1 mM ATP or 1 mM ADP and 5 μM FITC-labeled client where applicable. 10 μL of each sample was injected onto an Agilent AdvanceBio SEC 300 Å, 2.7 μm, 4.6 × 300 mm analytical SEC column pre-equilibrated with HPLC Buffer. Samples were run over the column at a flow rate of 0.35 mL/min. The protein standards and BiP were detected by absorbance at 280 nm, and FITC-labeled client was detected by absorbance at 495 nm.

### BiP Bulk FRET
The G518C/Y636C BiP variant was labeled with Alexa Fluor 555 (donor) and Alexa Fluor 647 (acceptor) as previously described[11]. Fluorescently labeled BiP was diluted to 50 nM in 60 mM HEPES pH 7.0, 50 mM KCl, 1 mg/mL BSA, and 1 mM DTT. For experiments with ADP, BiP was first incubated with 0.1 mM ATP and 0.1 mM $MgCl_2$ then 1 mM hexokinase-treated ADP was flushed in. ADP was hexokinase-treated with 0.005 units/μL hexokinase, 1 mM glucose, and 5 mM $MgCl_2$ at 37 °C for 1 h. FRET data was collected with a Fluoromax-4 spectrofluorometer with a donor excitation of 532 nm and donor and acceptor emission wavelengths of 567 nm and 668 nm respectively. Slit widths were set to 4 nm for both excitation and emission with an integration time of 0.5 s at 30 °C. FRET efficiency was calculated from the emission as acceptor/(acceptor + donor).

### Inhibitor docking
Docking analyses were performed with MatchMaker on UCSF Chimera[45] which was also used to generate all protein structures. For the Trap1 coiled-coil structure (PDB: 5F3K), Sung and colleagues suggest that the lid would be displaced upon the binding of nucleotide[21] and therefore the lid (residues 190-202) was removed.

### Grp94 crosslinking assay
Experimental conditions consist of 1 μM M86C Grp94 dimer, 50 mM KCl, 50 mM Tris pH 8, and 4% v/v DMSO. 5 μM BiP NBD and/or 50 μM inhibitor were added in relevant experiments. Crosslinking was initiated by 10 mM oxidized glutathione. Samples were then incubated at 25 °C for 2 h before quenching with a final concentration of 7.5 mM freshly made N-ethylmaleimide (NEM). Samples were boiled, stained, and loaded onto a 4–12% SDS-PAGE gel. ImageJ was used to quantify the amount of non-cross-linked (monomer) and cross-linked (dimer) Grp94[46].

### Reporting summary
Further information on research design is available in the Nature Portfolio Reporting Summary linked to this article.

### Data availability
The data that support this study. Source data are provided with this paper.

### Code availability
Code for analyzing smFRET images is available at https://github.com/gelles-brandeis/CoSMoS_Analysis.

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

## Acknowledgements

We thank members of the Street lab for helpful feedback. Research for this project was supported by NIH R01 GM115356 (T.O.S.), NIH R01 GM121384 and NIH R01 GM81648 (J.G.).

## Author contributions

T.P.A., E.E.D., J.H., B.H., and R.H. performed experiments and conceived the ideas in this paper. T.P.A. and B.H. conducted and analyzed the smFRET experiments. G.J. and L.J.F. provided assistance and feedback for smFRET experiments. T.P.A., E.E.D., J.H., G.J., L.J.F., and T.O.S wrote or edited the paper.

## Competing interests

The authors declare no competing interests.
