## [Peer Review file · Nature Communications]

Mechanism of client loading from BiP to Grp94 and its disruption by select inhibitors

Corresponding Author: Dr Timothy Street

Version 0:

Reviewer comments:

Reviewer #1

(Remarks to the Author)

Azam et al. have improved their manuscript and provided additional data to support their hypothesis on how drugs could interfere with the client loading from BiP to Grp94, which is a very important and noteworthy study.

Most of my questions have been well addressed, but there is one major point, which has to be addressed, before I can support publication: For a FRET trace to qualify as single-molecule FRET, at least a clear donor or acceptor bleaching step (better both) has to be visible, this is not shown for a single trace in this manuscript. In Fig. 2 of their J.Mol.Biol publication from 2019 this is nicely shown (Fig. 2), but this was five years ago and of course different data. Therefore, for this new manuscript it is crucial to shown at least for a few traces the data (donor, acceptor, direct excitation, FRET). Otherwise, it is impossible to judge the quality of the data. This is important for the readers, especially as some of the FRET efficiency data provided in the point-by-point-response shows unexpected peaks, or quite high noise, or even unexpected transitions (e.g. second trace in second row or second trace in third row). Even more, in my opinion, nowadays it is good scientific practice to make the traces available to the reviewers or deposit the data on a public repository (at least the basic data for the main text figures).

Minor points:

- 1) I still do not fully understand, why only the proposed delivery mechanism is possible. I understand now that the client binds well to BiP on its own. I also see that binding to Grp/Bip is much stronger (Fig. 6A). Therefore, why can the authors exclude that first BiP binds to Grp and then the client binds to the BiP-Grp complex?
- 2) Along the same line, I do not understand Fig. 5C: Does the upper panel show that under ADP conditions all client is bound to BiP (as there is only one peak)? How can this be the case here, when more client (5uM) is added than BiP (4uM)? Then, in the lower panel, there is more than 50% unbound BiP, but I cannot see the unbound client (which should be at least as much protein)?
- 3) Please show the SDS-PAGE for Suppl. Fig. 5 in the supplement.
- 4) Please show the ATP-dependent data from your bulk FRET assay, which should demonstrates the acceleration of Grp94 closure by NBD*.

Reviewer #2

(Remarks to the Author)

This revised manuscript by Azam et al. describes the differential effects of Hsp90 directed inhibitors on the structure and activity of Grp94 in the client loading state. As with the previous version of this manuscript, the work is technically excellent. The description and analysis of the experiments has now been improved. The inclusion of Ref 25, which was apparently not available when the original manuscript was submitted, supports the scientific premise of the work that BiP collaborates with Grp94 to mature at least a subset of Grp94 clients. Overall this is a substantial revision that is now acceptable for publication.

We thank both reviewers for their feedback on this manuscript.

Reviewer #1 (Remarks to the Author):

This manuscript examines the interaction between the ER chaperones BiP and Grp94. The focus is to understand how BiP binding transitions the Grp94 into a conformation that will accept the handoff of a client protein. The work makes use of the group's expertise in bulk and single molecule FRET to examine the effects of different BiP constructs, Grp94 mutations, and select Hsp90 inhibitor molecules on the Grp94 conformation and the formation of a BiP-Grp94-client protein complex.

The motivation for this study was the discovery by this same group a few years ago that the NBD of BiP stimulates the formation of a novel Grp94 state, determined by a characteristic FRET signature, termed C'. On the basis of a crosslink between Cys residues engineered in place of Met86 on Grp94, the group proposed (Ref 20) that the C' state resembled the crystal structure of a Trap1 NTD dimer that was stabilized by a coiled-coil structure formed by alpha helices at the two Trap1 N termini. This observation puts the C' model for Grp94 at odds with a competing structural model, termed the semi-closed state, that is derived from a cryoEM study of Hsp90 with Hsp70 and a steroid receptor client. Much of the manuscript deals with trying to decide between these two models of the C' state.

We thank the reviewer for the careful reading of this manuscript and of our previous work. However, our motivation for this study differs from the above description. Our motivation was the surprising discovery that some inhibitors disrupt the client loading state whereas others do not – the data shown in Figure 2. Hopefully the purpose of the inhibitor experiments and the conclusions drawn from these experiments are clarified in the revised manuscript and from our responses below.

I feel that the use of the Hsp90 inhibitors to probe the C' state is flawed in its interpretation. The authors find that inhibitors HSP990 and XL888 prevent the C' state, while AUY922 does not. On the basis of structures of Hsp90 with these inhibitors, the authors identified Hsp90 Phe138 (corresponding to Grp94 Phe199) as a residue that might distinguish between the semi closed model and the coiled coiled model. Mutating the Phe199 to Ala shows that HSP990 can now achieve the C' state in this mutant but not XL888, leading to the conclusion that the C' state is not the coiled coil model.

The data in Figure 2 shows that some inhibitors are compatible with the client loading conformation whereas other inhibitors are not, which is an important discovery that we wanted to understand in greater detail. The purpose of the mutations was to determine why AUY922 is compatible with the C' state while XL888 and HSP990 are not, as stated in the first sentence of the subsection on page 5:

“Rather than focus on the Grp94 conformations with intermediate FRET efficiency that become populated from HSP990 and XL888, we first wanted to understand why AUY922 is compatible with the Grp94 high FRET C' state whereas HSP990 and XL888 are not.”

The conclusions were at the end of the subsection on page 6:

“Collectively the above results suggest that sterics explain why some inhibitors allow BiP to push Grp94 into the loading state whereas other inhibitors block this change. Specifically, XL888 and HSP990 would clash with the nucleotide binding pocket in the loading state. In the case of HSP990, we conclude that the clash is primarily with F199.”

While it is correct to say our results are consistent with the predictions from the semi-closed structure and inconsistent with the predictions from coiled-coil structure, this was not the purpose of the inhibitor experiments, and was not the conclusion drawn from the experiments. We have revised the text in this section to enhance the clarity.

I think it is important to note that the data for this experiment is very well done. But there is a problem with the interpretation, which is that the actual effect of the HSP990 and XL888 inhibitors on the structure of Grp94 is unknown and may not correspond to the Hsp90 structures. There are numerous structures of Grp94 in the PDB that show large conformational changes upon inhibitor binding. Are any of these models compatible with the C' state? From what I can see, the only conclusion to be had from this set of experiments is that the C' state does not resemble the Trap1 coiled coil. The authors could strengthen their manuscript by modeling in the available Grp94 NTD structures to determine if any of these are compatible with the C' state and the Met86Cys mutant crosslink.

The manuscript already included such modelling. Supplemental Figure 13 shows a modelled NTD from a Grp94 structure in which the N-terminal alpha-helix is partially unfolded and found that it would be compatible with the Met86Cys crosslink in the semi-closed conformation. This was described in the Discussion section.

As it stands now, the authors have only invalidated a straw man model for the C' state and called into question whether the Met86Cys crosslink is a true test for this state.

We would like to re-emphasize that the conclusion of this section of the manuscript pertains to the effects of the inhibitors on the Grp94 conformation, rather than the possible use of inhibitors to differentiate between structural models. Nevertheless, in the spirit behind the above comment we performed an experiment. If the Met86Cys crosslink is a valid test of the C' state then we would expect comparable crosslinking in the absence of inhibitor (apo) and with the C'-compatible inhibitor AUY922. In contrast, we would expect decreased crosslinking the C'-compatible inhibitors HSP990 and XL888. This is indeed what we observe. This result shows that the Met86Cys crosslink continues to be a valid test of the Grp94 C' state and is consistent with our findings about inhibitor compatibility with the C' state. We have added this data as Supplemental Figure 5 and incorporated this finding into the Discussion section.

Supplemental Figure 5: Quantification of M86C Grp94 crosslinking as measured by integrating bands on a non-reducing SDS PAGE. Unless otherwise indicated, samples contain 5 μ M BiP NBD; in samples with inhibitor the concentration is 50 μ M. Crosslinking was quenched at 2 hours. Bar represents the average from two independent experiments.

A second caveat about the focus on Hsp90 inhibitors in this manuscript is the suggestion that these inhibitors may act by disrupting the loading conformation of Grp94/BiP and that future drug design efforts might benefit from targeting this conformation. Unfortunately, this suggestion is not supported by any experimental evidence and does not support the significance of this work. In particular, there is no evidence that disrupting the BiP/Grp94 interaction prevents the maturation of client proteins in vivo. Without this demonstration, suggesting that disrupting the loading state with inhibitors is speculative.

Disrupting the BiP/Grp94 interaction has indeed been established to disrupt client folding *in vivo*. Please see:

Amankwah et al. *Structural transitions modulate the chaperone activities of Grp94.* PNAS 2024 Mar;121(12) e2309326121. doi:10.1073/pnas.2309326121. PMID: 38483986.

The relevant findings are described on page 7 under the subheading: “Grp94 Collaborates with BiP to Fold Proteins In Vivo.”

We have added text highlighting this result.

The above criticisms aside, there are a couple of very strong points to this manuscript. The authors have developed a nice FRET system to probe the C' state, and have identified BiP fragments and mutants that show that BiP uses contacts at interface II to drive Grp94 into the C' state. Equally important is the excellent identification of an experimentally tractable fragment of the client IGF2.

We thank the reviewer for the positive comments.

A few specific comments are listed below, and with the absence of line numbers the first word of the relevant line is indicated:

p. 2, NTD. Please indicate that the crosslink was via Met86, which corresponds to

Met30 in Hsp90.

Done.

p. 4, The initial. I don't see how reference 20 indicates that inhibitors have variable effects on the Grp94 C' state.

Reference 20 describes the bulk FRET assay that we use, not a finding about inhibitors. The results with inhibitors are shown in Supplemental Figure 2.

p. 4, BiP. Chatty. Perhaps rewording would be appropriate.

Done.

p. 4, conformation. Chatty

Done.

p. 5, AUY922. Please give the Trap1 reference and PDB in the text.

p. 5, Hsp90a. Please give the Hsp90 reference and PDB in the text.

The pdb codes were in the caption of Figure 2 and the references were in the introduction.

p. 5, Figure 2. In the schematic, perhaps labeling the components of the composite models that arise from Trap and Hsp90 would add to the clarity.

This labeling was in Figure 1 where the coiled-coil region from Trap1 and interface II from Hsp90 are boxed out.

p. 5, Figure 2. In the structural models it would help if arrows were used to point out the clashes that the authors are referring to.

Done.

p. 5, Figure 2. The Phe and Leu residues should also be labeled.

The Figure 2 caption provided this information:

“Residues corresponding to Hsp90 α L107 and Hsp90 α F138 are shown in magenta and green respectively.”

p. 6, line 7. A more systematic survey of the PDB should be done to make this statement. There are over 300 structures of Hsp90 in the PDB and I doubt the authors are familiar with them all. For example, PDB 7ur3 also has the oddly positioned Phe138. Does this change their argument?

We thank the reviewer for this observation. This does not change the argument, which is that the flipped rotamer of Phe138 stands out as a rare feature of the nucleotide binding pocket. We have revised this section to enhance the clarity.

It is also worth noting that the experimental evidence for modeling Phe138 in the semi closed cryoEM structure into the odd conformation is modest and looks like it could also be modeled into the normal Phe138 conformation as well.

Despite the modest resolution of the nucleotide binding pocket in the semi-closed structure, the results with F199A in Figure 3 indicate that this residue plays a critical role in explaining why HSP990 is incompatible with the Grp94 loading state.

p. 6, para 2. I don't buy this argument. Superposition of PDB 4u93 on 7kw7 shows no obvious clash that would preclude adopting the C' conformation, if it does resemble the semi closed state. The authors would have to do smFRET on the 90/70 complex in order to confirm this interpretation.

The clash is between HSP990 on 4u93 and Phe138 on Hsp90. This was shown in the lower middle panel of Figure 2F. This clash is now marked with an arrow. When this clash is removed by the F199A mutation Grp94 can now adopt the C' conformation in the presence of HSP990, as shown in Figure 3C. This led to the conclusion that:

“In the case of HSP990, we conclude that the clash is primarily with F199.”

This conclusion does not require smFRET measurements on the cytosol 70/90 complex.

p. 6, population. Looking at the Grp94 structure it is surprising that the Phe199Ala mutation is stable.

We were also curious about the stability of Grp94 Phe199Ala, which is why we performed the controls in Supplemental Figure 7 which show that the Grp94 Phe199Ala variant is active, stimulated by the BiP NBD, and fully inhibited by HSP990, XL888, and AUY922.

p. 12, Figure 6C. The effect of the HSP990 and XL888 inhibitors on the formation of the loading complex is modest, especially compared to the BiP and Grp94 mutations and to the effect on the C' smFRET data presented in earlier Figures. This calls into question whether inhibitors of the loading state are really going to be a valid path to therapeutics.

Because HSP990 and XL888 were developed prior to any knowledge about the loading state it is unlikely that they represent the maximum possible disruption. This point was made at the end of the Results section:

“This suggests room for improvement in designing Hsp90 inhibitors that destabilize client loading complexes to an even greater extent than either XL888 or HSP990.”

p. 12, Here. Perhaps replacing “fundamental” with “functional” would be more accurate.

Done.

p. 12, state. Replace “cytosol” with “cytosolic.”

Done.

p. 13, disrupt. Add in AUY922 after “compatible inhibitor.”

Done.

p. 13, the Trap1. Reference 21 should be Ref. 20.

Thank you for spotting this error.

p. 13, conformation. What recent analysis are the authors referring to? Isn't that this manuscript?

Thank you for spotting this - we did not include the reference. The analysis is in a recent paper from the Kravats lab:

Amankwah et al. *Structural transitions modulate the chaperone activities of Grp94.*
PNAS 2024 Mar;121(12) e2309326121. doi:10.1073/pnas.2309326121. PMID: 38483986.

Reviewer #2 (Remarks to the Author):

Azam et al investigate the client loading from BiP to Grp94 and how drugs could interfere with this process. The authors claim that they found the mechanism of BiP driving Grp94 into the client loading state. In addition, they show how a selected group of inhibitors disrupts the client loading. Finally, they claim that this client loading mechanism is conserved between the Hsp70/Hsp90 system in the ER and the cytosol. Altogether, the authors use an impressive combination of smFRET, bulk FRET and biochemical methods. The detailed understanding and the new avenue for rational Hsp90 drug design would make this study very interesting for the readership of NSMB, but in my opinion several claims are not fully justified as detailed in the following. Therefore major revisions are required:

1) In Figure 2 the authors nicely show how smFRET data is necessary to clearly distinguish different (loading) states and that mean bulk FRET efficiencies are not sufficient to do so. Then, in Figure 3 (to justify steric clashes) most of the smFRET data is missing. In Fig. 3B,C the authors show the data for the F199A mutant, but what about the L163 mutant? Then, there is also only the data for one inhibitor (HSP990) shown, what about the other two inhibitors (XL888 and AUY922)? As all the constructs are available, this should not be too much work. This kind of quantitative data would (in my view) be necessary to justify the conclusions.

For reference, the constructs used for bulk FRET are different from ones in smFRET. The smFRET constructs have a C-terminal SNAP-tag that is biotinylated so that the protein can be tethered to the smFRET microscope slide. Details about smFRET constructs and labelling can be found in:

Huang, B., Friedman, L. J., Sun, M., Gelles, J. & Street, T. O. Conformational Cycling within the Closed State of Grp94, an Hsp90-Family Chaperone. *J Mol Biol* **431**, 3312–3323 (2019).

We constructed the new SNAP-tagged L163A Grp94 variant and performed smFRET measurements with the construct both alone, as well as with BiP and the inhibitors. We also collected new smFRET data with the F199A Grp94 variant in the presence of BiP with AUY922 and XL888. This new data is shown in Supplemental Figure 8.

As expected, Grp94_{L163A} in the absence of inhibitor and BiP results in the open state being populated and upon the addition of BiP, the C' state is populated (Supplemental Figure 8A). In the presence of BiP and AUY922, both Grp94 mutants remain C' compatible (Supplemental Figure 8B). Also as expected, no C' state is populated for Grp94_{L163A} in the presence of BiP and HSP990 (Supplemental Figure 8D). Interestingly, for Grp94_{L163A} in the presence of BiP and XL888/HSP990 and for Grp94_{F199A} in the presence of XL888, a new broad population of variable intermediate FRET efficiencies are observed (Supplemental Figure 8C and D). These new intermediate FRET populations were hidden in the bulk FRET results and so we thank the reviewer for the recommendation to perform these additional single molecule experiments. A small population of C' state is observed with Grp94_{L163A} in the presence of BiP and XL888 (Supplemental Figure 8C). Nonetheless, the only mutation that results in a complete shift to the C' state is for Grp94_{F199A} in the presence of BiP and HSP990. These results are consistent

with our previous conclusion that steric clashes explain why some inhibitors prevent Grp94 from accessing the C' state.

Supplemental Figure 8: Grp94_{L163A} and Grp94_{F199A} smFRET efficiency histograms. Results are shown for measurements in the absence of any ligand (Apo, **A**), and in the presence of 50µM AU922 (**B**), 50µM XL888 (**C**), 50µM HSP990 (**D**). $\langle E \rangle$ is the average FRET efficiency for the entire histogram. See Methods for experimental details.

2) In Figure 4E I would expect sigmoidal binding curves, e.g. as the authors have shown in one of their previous studies (Ref. 20, Fig. 5c). Why is the Fig. 4E so different? Why is the x-axis not shown on a logarithmic scale?

The data in Ref. 20, Fig. 5c was plotted on a logarithmic x-axis scale because the wild-type and cross-linked Grp94 have such different affinities that a logarithmic scale was needed to visualize both curves. The data in Fig 4E did not require a plotting on a logarithmic scale.

Please specify the “single-site binding equation”.

This was given in the Methods section:

“ K_D values were calculated using the single-site binding equation:

$$P = \frac{a[x]}{K_D + [x]} + c$$

Where P is the fluorescence polarization, a is the polarization amplitude, c is the polarization value in the absence of Grp94, and x is the concentration of Grp94.”

3) On page 10 the authors claim that the primary means of proIGF2₂₅₋₁₂₀ binding to Grp94 is via delivery from BiP. I cannot follow this argument. I understand that proIGF2₂₅₋₁₂₀ has low affinity to GRP94 on its own and to BiP on its own, but high

affinity to the Grp94/BiP complex. This does not tell anything about a delivery mechanism? What other evidence did I miss?

ProIGF2_25-120 binds well to BiP on its own. The binding between BiP and proIGF2_25-120 is shown the co-elution in Figure 5C and by FP in Supplemental Figure 11. This was summarized on page 9:

“Overall, we conclude that proIGF2₂₅₋₁₂₀ is a well-behaved monomeric client construct that can be stably trapped by BiP in the lid closed state, which makes proIGF2₂₅₋₁₂₀ well-suited for examining the client loading process and how it is impacted by inhibitors.”

Since BiP can stably bind the client but Grp94 does not, then the primary way in which Grp94 will receive the client would be from BiP binding the client and then delivering the client to Grp94. We have revised the wording to make this section more clear.

4) I do not see the evidence for the claim of a conserved mechanism. The authors make prediction on how to test this (“It will be interesting to see whether differences between AU922, HSP990, and XL888 are observed for client loading in the cytosol specific Hsp70/Hsp90 system.”), but they did not do these experiments.

We thank the reviewer for spotting this ambiguity in the writing. What we mean is that the client loading mechanism has specific conserved features between the Hsp70/Hsp90 systems in the ER and cytosol. In particular, we show that the Interface II, which was identified in the cytosol loading structure, plays crucial functional roles for BiP/Grp94 in:

- Forming the loading conformation (Figure 4B)
- Accelerating ATP-dependent Grp94 closure and ATP hydrolysis (Figure 4C&D)
- Stabilizing BiP/Grp94/client ternary complex (Figure 6C)

In contrast, it is not known whether the disruption of client loading by select inhibitors is a conserved feature. We have adjusted the text to clarify this point.

Minor points:

a) Figure 1: Please specify in the figure caption on which publications this model is based.

Done.

b) The authors use very often phrases like “might explain” or “indicates” or “would expect” or “could have”. Please write more clearly, what has been shown/ explained and what are full speculations.

We have revised the text in many places to make the conclusions more clearly articulated.

c) Please show a few smFRET traces or make them available for the referees. This is important to judge the quality of the traces and therefore FRET efficiencies.

We provided traces for the Grp94 smFRET system in previous publications. Please see Figure 2 and Supplemental Figure 3 in:

Huang, B., Friedman, L. J., Sun, M., Gelles, J. & Street, T. O. Conformational Cycling within the Closed State of Grp94, an Hsp90-Family Chaperone. *J Mol Biol* **431**, 3312–3323 (2019).

Here are some additional traces for the data in this manuscript.

d) The authors give SEMs for two measurements, this should not be done, because the sample size is far too small. You may e.g. give the error of one measurement and show the independent repeat in the supplement.

Done.

Reviewer #1 (Remarks to the Author)

Azam et al. have improved their manuscript and provided additional data to support their hypothesis on how drugs could interfere with the client loading from BiP to Grp94, which is a very important and noteworthy study.

Most of my questions have been well addressed, but there is one major point, which has to be addressed, before I can support publication: For a FRET trace to qualify as single-molecule FRET, at least a clear donor or acceptor bleaching step (better both) has to be visible, this is not shown for a single trace in this manuscript. In Fig. 2 of their J.Mol.Biol publication from 2019 this is nicely shown (Fig. 2), but this was five years ago and of course different data. Therefore, for this new manuscript it is crucial to shown at least for a few traces the data (donor, acceptor, direct excitation, FRET). Otherwise, it is impossible to judge the quality of the data. This is important for the readers, especially as some of the FRET efficiency data provided in the point-by-point-response shows unexpected peaks, or quite high noise, or even unexpected transitions (e.g. second trace in second row or second trace in third row). Even more, in my opinion, nowadays it is good scientific practice to make the traces available to the reviewers or deposit the data on a public repository (at least the basic data for the main text figures).

We thank the reviewer for the helpful feedback.

- We have now included the raw data for donor, acceptor, and direct acceptor excitation for representative traces in Supplemental Figure 4.

Supplemental Figure 4: Example FRET traces (left) and corresponding donor, acceptor, and direct excitation traces (right) of individual Grp94 dimers corresponding to the open conformation or intermediate FRET conformation. These data correspond to histograms in **Figure 2D**, in conditions with 8 μ M BiP NBD and 50 μ M XL888 or HSP990. Donor fluorescence is shown in green, acceptor fluorescence is shown in red, and direct acceptor excitation fluorescence is shown in black.

Minor points:

1) I still do not fully understand, why only the proposed delivery mechanism is possible. I understand now that the client binds well to BiP on its own. I also see that binding to Grp/Bip is much stronger (Fig. 6A). Therefore, why can the authors exclude that first BiP binds to Grp and then the client binds to the BiP-Grp complex?

- Please note that the magnitude of an FP value is an indication of the complex size or tumbling rate. So the larger FP values in Figure 6A do not show client binding to Grp94/BiP as much stronger, but rather that the BiP/client/Grp94 complex as larger than the BiP/client complex. The concentration-dependence determines an apparent affinity, and the values are comparable between BiP/client and BiP/client/Grp94. This was mentioned towards the end of page 11:

“The apparent affinity of proIGF₂₂₅₋₁₂₀ to BiP/Grp94 is comparable to BiP alone (compare red and green curves in **Figure 6A**), but this should be interpreted cautiously because the FP signal likely has contributions from proIGF₂₂₅₋₁₂₀ binding to both BiP and BiP/Grp94 over the experimental concentration range.”

Nevertheless, this is still a good question. A key reason why we can exclude a model in which BiP binds Grp94 first and then the client binds afterwards relates to the conditions in which BiP is able to bind to Grp94. On page 2 of the introduction, we discussed a previous finding with BiP and Grp94:

“When ATP is bound, BiP adopts a conformation in which the substrate-binding domain (SBD) is docked onto the NBD, opening a lid that allows clients to transiently bind the SBD²³. ATP hydrolysis by BiP causes the SBD to undock, which in many, but not all cases²⁴, results in stable lid closure over the bound client. Grp94 can directly bind to BiP when the SBD is undocked and the lid is closed, but not when BiP is in the ATP conformation¹¹. This mechanism enables BiP to deliver client proteins to Grp94.”

Under conditions in which ATP is present, BiP will be primarily in a conformation that is incapable of binding to Grp94. This is why we expect client binding to occur first to BiP, which causes the lid to close, and consequently allows BiP to bind to Grp94, as is shown in Figure 1. Under conditions in which no ATP is present it could be possible for BiP to bind to Grp94 first, but these are not biologically relevant conditions.

2) Along the same line, I do not understand Fig. 5C: Does the upper panel show that under ADP conditions all client is bound to BiP (as there is only one peak)? How can this be the case here, when more client (5 μ M) is added than BiP (4 μ M)? Then, in the lower panel, there is more than

50% unbound BiP, but I cannot see the unbound client (which should be at least as much protein)?

- Figure 5C just shows the peaks corresponding to BiP bound to client. You are correct that all client cannot be bound to BiP, and this excess client elutes later at 12 mins. Here is a chromatogram with the free client peaks.

3) Please show the SDS-PAGE for Suppl. Fig. 5 in the supplement.

- We replicated the crosslinking experiment and included the corresponding gel for the Supplemental Figure.

4) Please show the ATP-dependent data from your bulk FRET assay, which should demonstrate the acceleration of Grp94 closure by NBD*.

- Here is representative kinetic bulk FRET data showing that NBD* accelerates Grp94 closure while NBD-BiP₁₁₉₁ and NBD-BiP_{D187} do not.

Reviewer #2 (Remarks to the Author)

This revised manuscript by Azam et al. describes the differential effects of Hsp90 directed inhibitors on the structure and activity of Grp94 in the client loading state. As with the previous version of this manuscript, the work is technically excellent. The description and analysis of the experiments has now been improved. The inclusion of Ref 25, which was apparently not available when the original manuscript was submitted, supports the scientific premise of the work that BiP collaborates with Grp94 to mature at least a subset of Grp94 clients. Overall this is a substantial revision that is now acceptable for publication.

- We thank the reviewer for this feedback on our revised manuscript and its improvements.